# A Soil-Isolated *Streptomyces spororaveus* Species Produces a High-Molecular-Weight Antibiotic AF1 against Fungi and Gram-Positive Bacteria

**DOI:** 10.3390/antibiotics11050679

**Published:** 2022-05-18

**Authors:** Pu-Chieh Chang, Shao-Chung Liu, Ming-Chun Ho, Tzu-Wen Huang, Chih-Hung Huang

**Affiliations:** 1Department of Chemical Engineering & Biotechnology, Institute of Chemical Engineering, National Taipei University of Technology, Taipei 10608, Taiwan; t104679007@ntut.org.tw (P.-C.C.); shinning_maple@yahoo.com.tw (S.-C.L.); 2School of Pharmacy, Taipei Medical University, Taipei 11031, Taiwan; stannia@fda.gov.tw; 3Department of Microbiology and Immunology, School of Medicine, College of Medicine, Taipei Medical University, Taipei 11031, Taiwan; 4Graduate Institute of Medical Sciences, College of Medicine, Taipei Medical University, Taipei 11031, Taiwan

**Keywords:** *Streptomyces*, antifungal, antibacterial, PacBio, zone of inhibition

## Abstract

The overuse of antibiotics has resulted in the emergence of antibiotic resistance, not only in bacteria but also in fungi. *Streptomyces* are known to produce numerous secondary metabolites including clinically useful antibiotics. In this study, we screened for antibiotic-producing actinobacteria from soils in Taipei and discovered a *Streptomyces* strain SC26 that displayed antimicrobial activities against Gram-positive bacteria and fungi, but the compounds are heat-labile. Upon UV mutagenesis, a late-sporulation mutant SC263 was isolated with the same antibiotic spectrum but increased in thermostability. The nature of the antibiotic is not clear, but its activity was resistant to proteolytic, nucleolytic and pancreatic digestions, and was retained by the 100 kDa membrane during filtration. To gather more information on SC263, the genome was sequenced, which produced three contigs with a total of 8.2 Mb and was assigned to the species of *Streptomyces spororaveus* based on the average nucleotide identity to the reference species *S. spororaveus* NBRC 15456.

## 1. Introduction

The development of antimicrobial resistance in pathogenic bacteria and fungi has led to global health emergencies [1]. In both developed and developing countries, antibiotics are widely used in livestock to promote growth and to prevent infectious diseases [2,3]. As a result, meat consumed by humans and animals is often contaminated with the antibiotics resistance in the intestinal flora [4,5]. The annual deaths resulting from diseases caused by drug-resistant pathogens is estimated to be over 700,000 in more than 120 countries according to the World Health Organization [6]. However, the number of new drug applications has decreased significantly from 19 (1980–1984) down to just 6 (2010–2014) in the U.S. [7] The urge for new antibiotics or antifungal compounds from natural products is eminent.

Fluconazole- and azole-resistant strains of fungi are emerging [8,9]. Patients with immunosuppression are prone to be infected with multi-drug resistance *Candida* [10] and *Aspergillus* [11]. Nearly all clinical antifungal drugs exhibit a molecular weight lower than 2000 Da [12]. These commonly used drugs for antifungal infections, including amphotericin B (used for intravenous injections), flucytosine (used for oral and intravenous injections), and itraconazole (used for oral and intravenous injections), may exhibit severe side effects such as nephrotoxicity, hepatoxicity, bone marrow suppression, and heart failure, limiting the use of these drugs [13,14,15]. Drugs with less toxicity, such as natamycin produced from *Streptomyces* species, are considered relatively safe for topical application such as eyedrops, but is nearly insoluble in water with low bioavailability when orally used [16].

*Streptomyces* species, which are ubiquitous in fertile soil, are the major producers of natural products. They produce nearly 8000 natural products, accounting for approximately 45% of all from the microbial sources [17]. About 2/3 of clinically useful antibiotics are produced by *Streptomyces* [18], including many important drugs, such as streptomycin, neomycin, kanamycin, rapamycin, chloramphenicol, vancomycin, etc. [19]. This genus is predicted to possess a repertoire of approximately 100,000 antibiotics, of which only 3% has been discovered [20]. For decades, *Streptomyces* have been continuously explored for novel and useful antibiotics [21,22].

Here, we reported the isolation of a new *Streptomyces* strain SC26 from the mountains in Taipei. This isolate showed antimicrobial activity against six fungi and two Gram-positive bacteria but not Gram-negative bacteria. A late-sporulating mutant of SC26 was isolated (designated SC263) that exhibited increased thermostability of the antibiotic produced, named AF1. AF1 exhibited an apparent molecular weight larger than 100 kDa, which is unusual for antibiotics. The genomic sequence of SC263 was determined, which was classified to a known species *Streptomyces spororaveus*. A pairwise comparison between SC263 and *S. spororaveus* NBRC 15456, the only available whole genome sequences of *S. spororaveus* from the NCBI database, was carried out. AF1 has the potential for clinical applications, especially for antifungal applications.

## 2. Results and Discussion

### 2.1. Isolation of Antibiotic-Producing SC26

A total of six soil samples were screened for microbes on YM1 agar. Among colonies grown from these samples, isolate SC26 showed actinomycetes-like morphology, including grey spores (Figure 1A). Under a field-emission scanning electron microscope (FE-SEM), SC26 displayed mycelia and coiled spore chains (Figure 1B), characteristics of *Streptomyces*.

The culture supernatant of SC26 exhibited antibiotic activity against fungus *C. albicans* ATCC 10231 (Figure 2A). The 16S rRNA sequence of SC26 was determined. Phylogenetic analysis showed that the SC26 16S rRNA was clustered into the same clade as *Streptomyces spororaveus* NBRC 15456 and LMG 20313, *Streptomyces xanthophaeus* NBRC 12829 and NRRL B-5414, *Streptomyces nojiriensis* NBRC 13794 and LMG 20094 among the 100 *Streptomyces* species compared (Appendix A).

### 2.2. Isolation and Characterization of UV-Mutagenesis Derivatives

The antibiotic produced by SC26 (tentatively designated AF1) is heat labile, destroyed in 3–5 h at 65 °C (Figure 3A,B). UV mutagenesis of SC26 was conducted to search for strains producing relatively heat-stable AF1. Three mutant strains (SC261, SC262 and SC263) were chosen for their delayed sporulation (by 2 days compared to SC26). These mutants exhibited no difference in colony morphology and time course of AF1 production (Figure 2A) from SC26. We did not discover any non-producing strains among UV-irradiated derivatives.

Thermostability of AF1 produced by these mutants was conducted against *C. albicans* and *Staphylococcus aureus* ATCC 6538 (Figure 3). AF1 produced by SC262 exhibited similar heat lability as that produced by SC26, whereas AF1 produced by SC261 and SC263 were stable for up to 192 h at 25 °C against *C. albicans* and up to 144 h against *S. aureus*. SC263 was chosen for further detailed studies.

A pH stability assay was conducted on AF1 of SC26 and its derivatives. AF1 of SC26 was relatively unstable at pH 3 and pH 10 compared with those by its derivatives (Figure 2B). AF1 of SC263 was relatively stable from pH 2 to 12 but diminished at a pH of 13 to 14.

### 2.3. Antimicrobial Activities of AF1 from SC263

AF1 of SC263 exhibited the same antibiotic spectrum as SC26, i.e., effective against two fungi (*Aspergillus brasiliensis* ATCC 16404 and *C. albicans*) and two Gram-positive bacteria (*S. aureus* and *Bacillus subtilis* ATCC 6051), but not three Gram-negative bacteria (*Escherichia coli* ATCC 8739, *Salmonella* Typhimurium SL1344, and *Salmonella* Enteritidis ATCC 13076) (Figure 4A).

In addition, AF1 of SC263 also inhibited four other fungi, *Aspergillus niger*, *Candida tropicalis*, *Nannizzia gypsea*, and *Saccharomyces cerevisiae* (Figure 4B). The largest inhibition zone was found in *A. niger* with an average diameter of 32.0 mm, and the smallest was found in *N. gypsea* with 7.1 mm in diameter (Figure 4C).

These inhibition zones produced by AF1 on the two Gram-positive bacteria were opaque (Figure 4B), containing minute colonies which were able to grow when transferred to a new plate. In comparison, the inhibition zones in the fungal plates contained no viable cells. This suggests that the activity of AF1 against the two Gram-positive bacteria is bacteriostatic, not bactericidal.

### 2.4. Characterization of the SC263 Antibiotics by Membrane-Filtration

To estimate the size of the AF1 in SC263, we filtered the cultured broths through a 10-kDa and a 100-kDa pore-size membrane. The active compound was absent in the filtrate, but present in the >100 kDa fraction (designated ‘F100’; Figure 5A). F100, when lyophilized and reconstituted, exhibited the same antibiotic spectrum as the culture supernatant (Figure 5B). 

The antibiotic in SC26 broth yielded the same sizing results. The large apparent size of AF1 distinguishes it from the majority of known antibiotics, which are mostly small metabolites [23]. Most known antifungal antibiotics are below the molecular weight of 2000 Da [12].

Many large antifungal natural products are proteins [24,25]. Treatment of the SC263 broth and F100 with either proteinase K or pronase E did not eliminate antibiotic activity (Figure 6), suggesting that AF1 is not a protein. Treatment with DNaseI, RNaseA, or pancreatin (which composes of amylase, trypsin, lipase, ribonuclease, and protease) also did not extinguish the antibiotic activity of AF1 (data not shown).

### 2.5. Genomic Sequence of SC263

To taxonomically identify SC263 and to investigate the secondary metabolic repertoire of SC263, the whole genomic sequence of SC263 was determined. Three contigs, Ctg1 (7,201,269 bp), Ctg2 (996,243 bp), and Ctg3 (20,182 bp), were produced (Appendix A). The overall GC content is 71.92%.

A total of 7370 genes and 137 pseudogenes were annotated. Then, 91 RNAs, including 7 rRNAs, 73 tRNAs, and 3 non-coding RNAs were predicted. An origin of chromosome replication *oriC* was centrally located between the *dnaA* and *dnaN* genes on Ctg1 (position 3,676,752–3,677,627) [26,27,28].

Four genomes in the same 16S rRNA clade were used for pairwise comparison with SC263. The average nucleotide identity (ANI) [29] values against *S. spororaveus* NBRC 15456, *S. nojiriensis* NBRC 13794, *S. xanthophaeus* NBRC 12829 and NRRL B-5414 were 99.57, 92.88, 86.57, 86.47, respectively. We therefore classified SC263 as *Streptomyces spororaveus*. Genome-wide comparison shows relatively high synteny between SC263 and NBRC 15456.

### 2.6. Biosynthetic Gene Clusters in SC263

To assess the biosynthetic potential of secondary metabolites in SC263, we used antiSMASH webserver [30] to predict possible biosynthetic gene clusters (BGCs) on its genome. Ctg1 contained 21 predicted BGCs, Ctg2 contained 7, and Ctg3 contained none (Appendix A). None of the predicted secondary metabolites produced by these BGC are larger than 2000 Da.

The BGCs on the SC263 genome showed high similarity with those on the NBRC 15456 genomes (Figure 7). Compared to the NBRC 15456 genome, the SC263 genome contained three additional BGCs: meoabyssomicin/abyssomicin (Region 1–12, Appendix A), ectoine (Region 1–19), and lanthipeptide-class-I (Region 1–20).

Interestingly, another closely related strain, *S. spororaveus* RDS28, which was isolated in Saudi Arabia, also exhibits an antifungal activity to many plant pathogens [31]. The nature of this antibiotic (s) is unknown.

## 3. Materials and Methods

### 3.1. Soil Sample Collection and Isolation of SC26

Soil samples were collected from the mountains at Wenshan District in Taipei and stored in sterile bags at 4 °C until use. The samples were sieved through 1 mm metal mesh, suspended in sterile water, serially diluted, plated on YM1 agar (0.4% yeast extract, 1% malt extract), and incubated at 30 °C for 5 to 7 days. Individual colonies that produced spores were picked and streaked onto YM1 agar. Spores from these isolates were collected, suspended in sterile 20% glycerol, and stored at −80 °C.

### 3.2. Morphology under Electron Microscope

Spore chains of SC263 were collected from a 5-day culture growing on a cover slide on YM1 agar, fixed with liquid nitrogen, coated with gold particle, and observed under a Thermal Field Emission Scanning Electron Microscope (FE-SEM) JSM-7610F (JOEL Ltd., Tokyo, Japan).

### 3.3. Phylogenetic Analysis of 16S rRNA

Full-length 16S rRNA sequences was amplified by PCR using primers 27F and reverse primer 1492R [32] and subjected to Sanger sequencing. The 16S rRNA sequence of SC26 was compared to the database on BLASTN webserver [33] at National Center for Biotechnology Information (NCBI). One hundred top hit sequences were imported to MEGA11 for alignment and phylogenetic tree construction [34,35].

### 3.4. Microbial Strains and Culture Conditions

The fungi used in antibiotic tests were *Aspergillus niger* ATCC 16878, *Aspergillus brasiliensis* ATCC 16404, *Candida albicans* ATCC 10231, *Candida tropicalis* ATCC 13803, *Nannizzia gypsea* ATCC 14683, and *Saccharomyces cerevisiae* BCRC 22286. The two Gram-positive bacteria used were *Bacillus subtilis* ATCC 6051 and *Staphylococcus aureus* ATCC 6538. The three Gram-negative bacteria used were *Escherichia coli* ATCC 8739, *Salmonella* Typhimurium SL1344, and *Salmonella* Enteritidis ATCC 13076.

The test fungi were propagated on sabouraud dextrose agar (SDA). The test bacteria were propagated tryptone soy agar (TSA). All bacteria and fungi were incubated at 30 °C.

*Streptomyces* was cultured on solid YM1 agar and incubated at 30 °C, or shaken in liquid YM1 broth at 30 °C.

### 3.5. Cross Streak Screening for Antibiotic Activity

Actinomycete-like soil isolates were streaked on YM1 agar and incubated for 24 h at 30 °C. One microliter of the test bacteria or fungi containing 10^3^ colony-formation-unit (CFU) was spotted perpendicularly (Figure 4A), and incubation was continued for four more days [36].

### 3.6. Zone of Inhibition Assays for Antibiotic Activities 

All inoculants containing 10^5^ CFU each (except for *A. brasiliensis*, *A. niger*, and *N. gypsea*, which contained 10^6^ each) were spread on YM1 agar [37]. Then, 10 µL of liquid containing the antibiotic (AF1) was spotted. For culture supernatant, the culture was grown at 30 °C for 24 h and centrifuged at 12,000× *g* for 5 min. The supernatant was collected and used for inhibition assays. For lyophilized powder, 0.1 g was dissolved in 1 mL sterile water. The plates were incubated at 30 °C for 5 days.

### 3.7. Membrane-Filtration Concentration of SC263

Next, 10 kDa and 100 kDa Centrifugal Filter Unit (AMICON) were used to identify the molecular weight of AF1. Then, 2 L of SC263 culture supernatant was filtered through a 100 kDa mPES membrane (#D04-100 kDa, Repligen-Spectrumlabs) until the volume was reduced to 500 mL. The filtrated and the retained fractions were tested for antibiotic activities.

### 3.8. UV Mutagenesis of SC26

UV mutagenesis of *Streptomyces* was performed according to Kumar [38]. The surviving colonies were streaked onto a new plate and selected for late sporulating mutants.

### 3.9. Treatment with Lytic Enzymes

For cultured supernatant or F100 of SC263, proteinase K and pronase E were added to a concentration of 0.04 mg/mL and incubated at 37 °C for an hour. RNaseA or DNaseI were treated at 1 mg/mL and incubated at 37 °C for 0.5 h. Pancreatin (Sigma-Aldrich) from porcine pancreas, which composed of amylase, trypsin, lipase, ribonuclease and protease, was added at 10 mg/mL 37 °C for an hour.

### 3.10. Genomic DNA Purification

SC263 was grown in YM1 and incubated at 30 °C for 3 days. The mycelia were harvested by centrifugation and washed in 10% sucrose. Genomic DNA was extracted according to Kutchma et al. [39]. Purified genomic DNA was stored in TE buffer (10 mM Tris-HCl, 0.1 mM EDTA, pH 8.0) at −20 °C.

### 3.11. Whole-Genome Sequencing

Genomic DNA was processed according to the assay protocol of PacBio CLR. The sequencing format is the SMRTcell sequencing via Sequel IIe platform. The sequence reads were assembled using SMRT Link (version 10.1.0.119588). The resulting three contigs achieved a QV score at least 91.39. Contigs with a score lower than 90 were removed.

### 3.12. Genomic Analysis

Annotation of the SC263 genome was conducted using NCBI Prokaryotic Genome Annotation Pipeline (PGAP), and the results were deposited to the NCBI database with an accession number of JAKGSF000000000. Average nucleotide identity (ANI) was conducted using OrthoANI [29]. Genome-wide similarity comparison was conducted using Easyfig [40]. The presence of biosynthetic gene clusters was mined using antiSMASH 6.0 [30].

## 4. Conclusions

*S. spororaveus* SC26 was isolated at Wenshan District in Taipei, which produces an antibiotic (AF1) against six fungi and two Gram-positive bacteria. From SC26, a mutant SC263 was isolated, which exhibited the increased thermostability of AF1. AF1 produced by these strains is unusual in its apparent molecular weight, its dual antifungal and antibacterial activities. AF1 is resistant to different lytic enzyme digestion. Treatment with proteases, including pronase E and proteinase K, nucleases including RNaseA and DNaseI, pancreatin from porcine pancreas including amylase, trypsin, lipase, ribonuclease and protease, did not destroy AF1. These results suggest that AF1 is not a protein, nucleic acid, polysaccharide, or lipid.

The genomic sequences of SC263 contained 28 predicted BGCs, none of which appeared to be involved in the production of secondary metabolites larger than 2000 Da. Antibiotics with very high molecular weights are difficult to penetrate the skin barrier and are potentially safer for topical administration [41]. The identity of the antibiotic produced by SC26 and SC263 is being actively pursued.

## Figures and Tables

**Figure 1 antibiotics-11-00679-f001:**
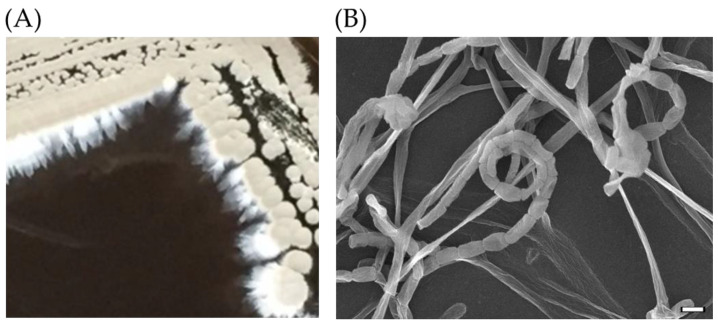
Morphology of SC26. (**A**) Colony morphology of SC26 on YM1 agar after incubation at 30 °C for 6 days. (**B**) Spore chains of SC26 under field-emission scanning electron microscope (FE-SEM). The bar indicates 1 μm.

**Figure 2 antibiotics-11-00679-f002:**
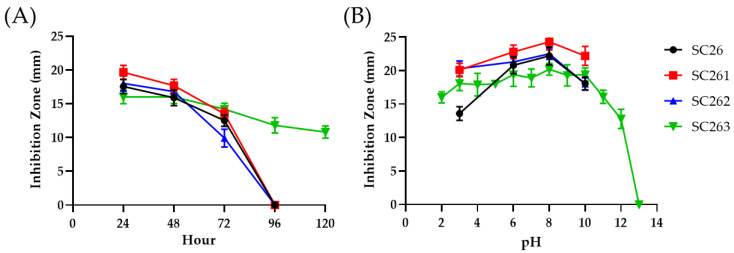
Antifungal characteristics of SC26 and its derivatives. The antifungal activity of SC26 and its derivatives was examined using different fermentation hours until day 5 (**A**) and treated under different pH conditions from 24 h cultured broth (**B**). All inhibition zones against *C. albicans* were measured.

**Figure 3 antibiotics-11-00679-f003:**
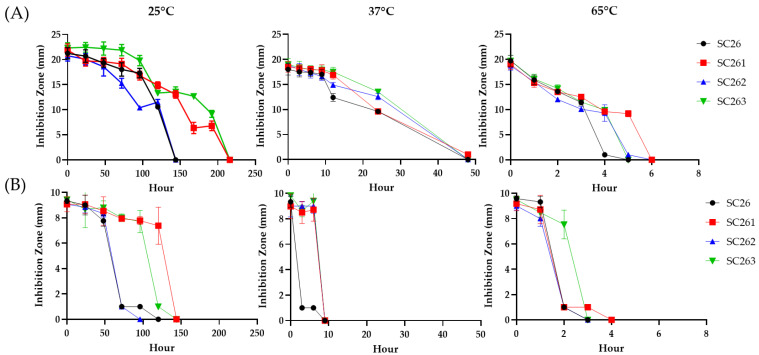
Temperature stability of bioactive compounds from SC26 and its derivatives. Overnight cultured broths from SC26 and its three derivatives (SC261, SC262 and SC263) were stored at 25 °C, 37 °C and 65 °C. Antimicrobial activity against *C. albicans* (**A**) and *S. aureus* (**B**) was measured at different time points.

**Figure 4 antibiotics-11-00679-f004:**
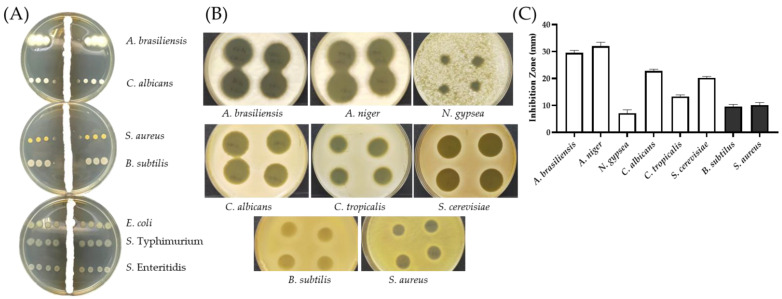
Antibacterial and antifungal activities of SC263. (**A**) Antibiotic activity of SC263 using cross-streak method for a 6-day incubation. An aliquot of 1 μL of bacterial suspension containing 10^3^ CFU were spotted cross the central line of SC263. (**B**) 10 μL of SC263 culture suspension was spotted in quadruplicates on lawns of test fungi and bacteria. *A. brasiliensis*, *A. niger*, and *N. gypsea*, was seeded with 10^6^ CFU each plate. The rest was seeded with 10^5^ CFU. (**C**) The diameters of inhibition zones were measured in triplicate. The error bars are calculated based on the quadruplicate zones for each tested microbe. White columns indicate fungi and black as Gram-positive bacteria.

**Figure 5 antibiotics-11-00679-f005:**
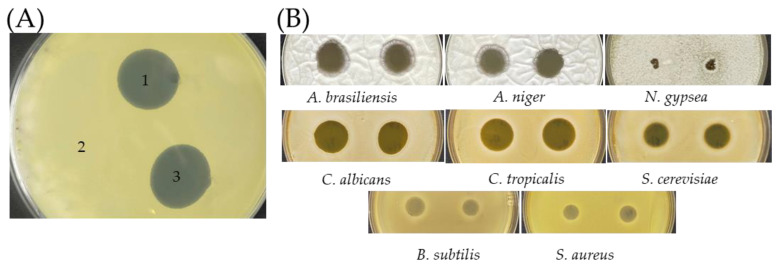
Antibiotic activities of SC263 membrane filtrate. (**A**) Different fractions of SC263 cultured broth were tested against *C. albicans*. 1: culture broth; 2: filtrate < 100 kDa; 3: compounds > 100 kDa (F100 Fraction). (**B**) Reconstituted lyophilized powder was tested for zone of inhibition in duplicate spots against different fungi and Gram-positive bacteria.

**Figure 6 antibiotics-11-00679-f006:**
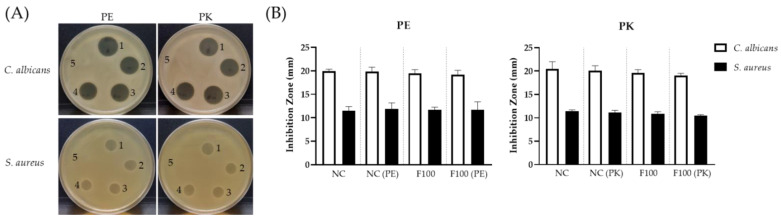
Protease Treatment of AF1 and F100 from SC263. (**A**) Antibiotic activities of AF1 and F100 individually treated with pronase E (PE) or proteinase K (PK) were examined. Numbers in this image indicate different treatment conditions as below. 1: AF1. 2: F100. 3: AF1 treated with PE or PK. 4: F100 treated with PE or PK. 5: PE or PK in sterile water. (**B**) The diameters of inhibition zones were measured.

**Figure 7 antibiotics-11-00679-f007:**
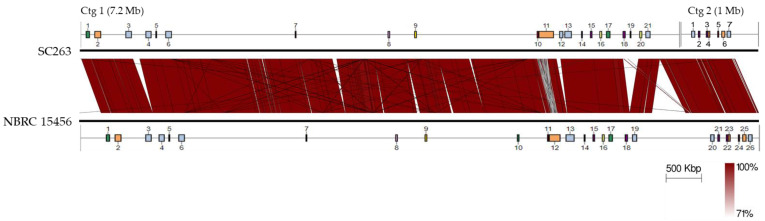
Genome comparison between SC263 and *Streptomyces spororaveus* NBRC 15456. Biosynthetic gene clusters (BGCs) were predicted using antiSMASH 6.0 webserver and numbered by the order of these two strains in the top and bottom lines, respectively. Genome comparison between these two strains was generated using EASYFIG v. 2.2.5 and placed in the center, where dark red displayed in 100% nucleotide identity and the color was gradually toward white with the percentage of identity.

## Data Availability

Not applicable.

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
