# Peer review of "A Soil-Isolated Streptomyces spororaveus Species Produces a High-Molecular-Weight Antibiotic AF1 against Fungi and Gram-Positive Bacteria"

_antibiotics, 2022, doi:10.3390/antibiotics11050679_

Round 1

Reviewer 1 Report

In this article, the authors isolate and characterize a new strain belonging to the Streptomyces genus (SC26). Initially, the authors perform a morphological and phylogenetic characterization (rRNA 16S) followed by UV irradiation and isolation of three mutant strains (SC261, SC262, SC263). The mutants were characterized for their ability to produce antifungal molecules. The molecule (or mixture?) of interest was treated at different pH and temperatures to examine the stability of the compound. A more in-depth analysis of antimicrobial activity was subsequently performed. This analysis also included filtration of the medium with 10-kDa and a 100-kDa pore-size membrane. Finally, the authors sequenced and analyzed the genome of streptomyces SC26, identifying the clusters involved in secondary metabolism. The study is simple but well, and I have only a few observations. 1. First, there is no standard deviation in the data shown in Figures 2, 3 and 6. Were the experiments performed in replicate? Please, define this point. 2. The authors find no possible explanation or hypothesis about the type of molecules of this size and similar activity. This point is very important, and the authors could discuss it in a dedicated section (discussion?). 3. It is important to understand the nature of the compound (s) that give this activity. The authors point out that it is not a protein and that it is probably not a secondary metabolite. However, it would be important to understand what else it can be. For this reason, controls should be carried out. For example, using extractions and/or digestion with other enzymes. 4. It would be interesting to see if there are strains among the UV mutants that do not produce antibiotics. Then the genome of the mutants could be analyzed to identify which genes are involved in the described antimicrobial activity. This approach could greatly improve the study. However, if the authors feel they are going to elaborate on this in subsequent studies, they can simply describe this point during the discussion. 5. As a tip: the authors call the compound AF1. However, antimicrobial tests cannot tell if there is only one molecule in the broth. For this, it would be necessary to use an HPLC (or another analytical method). I suggest changing the text by not talking about AF1 as if it were a single molecule.

Reviewer 2 Report

antibiotics-1659614-review

Specifically, please see my suggestion below based on the individual section of the manuscript:

INTRODUCTION:

  • Overall, the entire background/introduction part needs additional work and also lacks citations. A lot more citations are required throughout the introduction to substantiate the importance of this work. For something like this which has a global significance (paras 1 and 2), statements should not be supported by just a citation or two.
  • Line 51 – change to antimicrobial
  • Lines 50-59 – This is more of a repeat of the abstract. Authors should list here the goals of the study instead of the results as listed in this paragraph.
  • Results and discussions, line 63, AND methods line 191 - Location of from where this isolate was obtained needs some more detail. All I see is the authors say ‘from the mountain areas in northern Taiwan’ ; this is not very specific. Authors need to list a lot more specific detail on the actual locations from where the samples were collected. Addition of a map of the sites would also be very helpful here.
  • Methods line 230 – is there a specific standard by which these zones of inhibitions are calculated? How is it determined whether the antibiotic is inhibiting growth? Is there a standard in the zones available to determine this scientifically? For example, antibiotic resistance is determined based off the CSL standards, are there such standards for determining this same thing in a newly discovered antibiotic? If this is correct, then the authors need to list those standards here.
  • Figure S1: I see that the main node is supported by a bootstrap of 57. For a node that important in determining the structure of the tree, this should be at least >70.

Round 2

Reviewer 1 Report

The authors responded to my criticisms. The article is ready to be published.